# Temperature Effects in an Acousto-Optic Modulator of Terahertz Radiation Based on Liquefied SF_6_ Gas

**DOI:** 10.3390/ma14195519

**Published:** 2021-09-23

**Authors:** Pavel A. Nikitin, Vasily V. Gerasimov, Ildus S. Khasanov

**Affiliations:** 1Scientific and Technological Centre of Unique Instrumentation RAS, 117342 Moscow, Russia; khasanov@ntcup.ru; 2Department of Physics, Novosibirsk State University, 630090 Novosibirsk, Russia; v.v.gerasimov3@gmail.com; 3Budker Institute of Nuclear Physics SB RAS, 630090 Novosibirsk, Russia

**Keywords:** acousto-optic interaction, terahertz radiation, diffraction, liquefied inert gas

## Abstract

The acousto-optic (AO) diffraction of terahertz (THz) radiation in liquefied sulfur hexafluoride (SF${}_{6}$) was investigated in various temperature regimes. It was found that with the increase in the temperature from +10 to +23 ${}^{{}^{\circ}}$C, the efficiency of the AO diffraction became one order higher at the same amplitude of the driving electrical signal. At the same time, the efficiency of the AO diffraction per 1 W of the sound power as well as the angular bandwidth of the efficient AO interaction were temperature independent within the measurement error. Increase of the resonant sound frequency with decreasing temperature and strong narrowing of the sound frequency bandwidth of the efficient AO interaction were detected.

## 1. Introduction

Acousto-optical (AO) devices have found wide application in optical information processing in real time and are actively used in fiber-optics, tomography, microscopy, and astronomy [1,2,3,4]. They have proven to be effective in the ultraviolet, visible, and infrared spectral ranges. The AO interaction is based on the photo-elastic effect, due to which the sound wave forms a phase diffraction grating in the medium. Birefringent single crystals usually serve as the interaction medium in the above mentioned ranges [5]. Most of the modern AO devices operate under normal conditions and provide efficiency close to 100%. The development of efficient AO filters and deflectors requires the use of birefringent crystals. This makes it possible to improve the characteristics of these devices by several orders of magnitude in comparison with analogs based on optically isotropic media [6,7]. Unfortunately, in the terahertz (THz) range, birefringent crystals are practically opaque, while optically isotropic crystals (for example, germanium) feature a lower value of the AO figure of merit [8]. Therefore, studies of the AO effect in the THz range have not been carried out for many years and were considered unpromising. However, the growing interest in THz radiation necessitated search for a suitable medium for AO interaction. It was found that inert gases (for example, Xe or Kr) can be such a medium since they are transparent to THz radiation and have good AO properties [9]. Unfortunately, ultrasonic attenuation is large in the gases even under high pressure. In the work [9] it was shown that the ultrasound attenuation in the liquid phase of CF6 is less than in the gaseous one. For this reason, experiments were carried out in work [9] with various inert gases in the liquid phase and the AO diffraction efficiency of about 80% was obtained. It was found that the best suitable medium was liquefied sulfur hexafluoride (SF${}_{6}$), the density of which is about 1.5 times that of water, whereas its sound velocity is only about 300 m/s. Due to this rare combination of properties, liquefied SF${}_{6}$ features an AO figure of merit one order higher than that of the best birefringent crystal (paratellurite, TeO${}_{2}$) used in the visible range [10]. Liquefied SF${}_{6}$ is an optically isotropic medium making it possible to manufacture only commercial AO modulators of powerful THz radiation. The prime cost of such modulators is low and is determined mainly by the manufacturing cost of a cuvette for liquefied gas. The advantages of AO modulators of THz radiation are: (1) low power consumption (several tens of watts); (2) high operating speed (about 10 μs); (3) spatial separation of the diffracted radiation beam from the transmitted one.

Research in work [9] was carried out at the temperature of +13 ${}^{{}^{\circ}}$C and +14 ${}^{{}^{\circ}}$C. It was found that the temperature increase of only 1 degree leads to the diffraction efficiency falling about three times at the same amplitude of the electrical signal. The author cannot provide an explanation of the effect discovered. The experiment we performed at a temperature of +24 ${}^{{}^{\circ}}$C gave only a qualitative confirmation of the model, which takes into account the effect of temperature on the physical properties of SF${}_{6}$ [11]. We are confident that this discrepancy is due to the fact that the model does not take into account the structure of the sound field. If we assume that the ultrasound transducer acts like a piston, modeling of the structure of the sound beam can be performed even on low-power computers, using the expansion of the solution into Gaussian functions with known weight factors [12].

Note that because of the ultrasound attenuation in SF${}_{6}$, it is necessary to use low frequencies of ultrasound. In works [9,11] it was about 300 kHz. The choice of the frequency of 300 kHz is due to the need to deflect the diffracted beam at an angle much larger than the divergence angle of this beam, which is about 1 degree for a beam diameter of 10 mm and a wavelength of 130 μm. At the ultrasound frequency of 300 kHz, the deflection angle is about 8 degrees, which is sufficient for its spatial separation from the transmitted radiation beam. The use of higher frequencies is not advisable because of the high attenuation of ultrasound, which is proportional to the square of the frequency. The resonant frequency $F_{0}$ is determined by the thickness *h* of the sound transducer and sound velocity *V*: $F_{0}=2h/V$. Accordingly, the thickness of the transducer (h=6 mm at $F_{0}=300$ kHz) was comparable to its width, which was chosen close to the diameter of the radiation beam and was about 10 mm. For this reason, complex types of oscillations arise in the ultrasound transducer, and the distribution of the ultrasound amplitude on the surface of the transducer cannot be considered uniform. Modeling the oscillations of a rectangular transducer with its thickness close to its width is a difficult task, and due to the difference in the technology of manufacturing the transducers, the result of such modeling may differ significantly from the real state of affairs. We assume that the difference in the results of works [9,11] is associated with the use of different transducers. The motivation of the work is to obtain correct results. We are confident that it is necessary to carry out a series of experiments at different temperatures, but using the same ultrasound transducer. Therefore, in this work, we focused our efforts on the experimental study of temperature effects in AO modulators of THz radiation.

## 2. Materials and Methods

We used radiation of the Novosibirsk free electron laser 1 (FEL) with the wavelength λ=130μm, the polarization of which was controlled by wire polarizer 2, and the intensity was set by calibrated attenuators (see Figure 1). The radiation beam was incident on the center of the input optical window of the AO cell at the distance l=5 cm from the sound transducer. For observation of the diffracted radiation, the AO cell 4 was tilted at the Bragg angle, and the signal (modulated at a frequency of 10 Hz) from generator of electrical signals 5 was applied to the ultrasound transducer via amplifier 6. At a distance of about 30 cm after the AO cell, lens 7 was located, focusing the radiation into the receiver, Golay cell 8. The signal from the receiver was isolated from the background noise using lock-in amplifier 9.

At the temperature of about +20 ${}^{{}^{\circ}}$C, SF${}_{6}$ gas liquefies at a pressure of about 20 bar. Therefore, the AO cell body was built from high strength steel in the form of a cylinder (see Figure 2). A detailed description of the AO cell can be found in work [11]. We present only some of the modernizations of the AO cell during the preparation for this work. To generate a more uniform sound beam, we used a sound transducer (d=14 mm wide and L=80 mm long) made of piezoceramics PZT-19. To prevent the diffraction of THz radiation on the reflected sound wave, we disposed a duralumin plate (at a distance of about 10 cm) at an appropriate angle opposite the transducer. A closed loop water cooling system was used. Water came from a tank with a mixture of water with ice, then it passed between the double concentric walls of the AO cell, and returned back to the tank. Dry nitrogen blowing prevented water condensation on the outside of the optical windows.

The optimal turn angle of the AO cell and the optimal frequency of ultrasound were determined as a result of sequential adjustment of the ultrasound frequency and the angle of incidence of THz radiation on the AO cell to achieve the maximum intensity $I_{1}$ of the diffracted radiation. In this mode, the dependence of the intensity $I_{1}$ on the square of the amplitude *U* of the driving electrical signal was investigated. The operating point was chosen on the linear section of the obtained dependence. The transducer resonant frequency varies with temperature. Therefore, at a given temperature, we determined the optimal frequency at which the AO diffraction efficiency reached its maximum value. Then the dependence of the intensity $I_{1}$ on the angle θ of the AO cell turn was measured. Further, the dependence of the intensity $I_{1}$ on the frequency *F* of the electrical signal was recorded. The described cycle of experiments was carried out for different temperature regimes.

To interpret and compare the results, as a measure of the efficiency of the AO diffraction, we used the ratio of the intensity $I_{1}$ of the diffracted radiation to the intensity $I_{0}$ of the radiation transmitted through the AO cell. As a result, the diffraction efficiency ξ and diffraction efficiency $\xi_{\mathrm{norm}}$ per 1 W of the electric power $P_{\mathrm{el}}$ were determined [13]:(1)$$\xi =\frac{I_{1}}{I_{0}},\;\;\;\;\xi_{\mathrm{norm}}=\frac{\xi }{P_{\mathrm{el}}},$$
(2)Pel=U22|Z|cosϕ=U22|Z|Re(Z)|Z|=12U2Re(Z)|Z|2,
where cosϕ is the so-called power factor, equal to the ratio of the instantaneous real power used by an electrical load to the apparent power running through the circuit; |Z| and Re(Z) are the absolute value and real part, respectively, of the frequency dependent impedance measured with a vector network analyzer.

Since the electrical impedance *Z* of the ultrasound transducer depends on the frequency *F*, the amplitude *U* of the electrical signal changed together with the frequency. This can lead to asymmetry in the frequency dependence of the diffraction efficiency ξ(F), while the dependence $\xi_{\mathrm{norm}}\left(F\right)$ remains symmetric. Moreover, it is inappropriately to compare the diffraction efficiency ξ at different temperatures *t* and the same amplitude *U* of the electrical signal, because the impedance *Z* is a temperature dependent physical quantity [14]. Instead, it is necessary to use the diffraction efficiency per 1 W of electric power, i.e., $\xi_{\mathrm{norm}}\left(F\right)$.

The experimental value of the diffraction efficiency ξ was compared with the one predicted by our model [11]:(3)$$\xi =\frac{\pi^{2}}{2\lambda^{2}}\frac{M_{2}P_{a}}{d}L\exp\left(-\alpha_{s}l\right),$$
where $M_{2}$ is the AO figure of merit; $P_{a}$ is the acoustic power, which is usually considered equal to the input electric power $P_{\mathrm{el}}$; *d* and *L* are the width and length of the sound transducer; $\alpha_{s}$ is the sound power attenuation coefficient; *l* is the distance from the sound transducer at which the THz beam traveled.

All values of the parameters of the experimental setup required for calculations are given in Table 1.

As one can see from Equations (Equation 2) and (Equation 3), the diffraction efficiency ξ is proportional to the input electric power $P_{\mathrm{el}}$ and the square of the voltage amplitude *U*:(4)$$\xi =k_{P}P_{\mathrm{el}}=k_{U}U^{2},$$
where the factors $k_{P}$ and $k_{U}$ can be determined using the least square method (LSM).

The AO figure of merit $M_{2}$ and the refractive index *n* of the liquefied gas were calculated using the Lorentz–Lorenz equation [9,15]:(5)$$M_{2}={\left[\frac{\left(n^{2}-1\right)\left(n^{2}+2\right)}{6n}\right]}^{2}\frac{4}{\rho V^{3}},\;\;n=\sqrt{1+\frac{2A\rho }{1-2A\rho /3}},$$
where ρ is the density; $A=1.64\cdot 10^{-4}$ m${}^{3}$/kg is a factor proportional to the mean polarizability of the gas molecules [11].

The density ρ of the liquefied gas was represented by a rational function [16]:(6)$$\rho =\frac{a_{1}+a_{2}t+a_{3}t^{2}+a_{4}t^{3}}{1+a_{5}t},$$
where the coefficients $a_{n}$ depend on the pressure *p*; *t* is in (${}^{{}^{\circ}}$C) and ρ is in (kg/m${}^{3}$). Numerical values of $a_{n}$ can be found in [16] for the temperature range from 0 to +50 ${}^{{}^{\circ}}$C and pressures from 12 to 200 bar. The values of $a_{n}$ used in our work can be found in Table 2.

The sound power attenuation coefficient $\alpha_{s}$ was calculated using the relation obtained in work [11] by the LSM for the temperature range from +9 to +30 ${}^{{}^{\circ}}$C:(7)$$\alpha_{s}={\left(\frac{F}{F_{0}}\right)}^{2}\cdot \left[2466+4.81t^{1.875}\right]\cdot 10^{-4},$$
where $\alpha_{s}$ is in (cm${}^{-1}$), *t* is in (${}^{{}^{\circ}}$C), and $F_{0}=300$ kHz.

Data on the sound velocity *V* in liquefied SF${}_{6}$ gas for a wide range of temperatures (from −40 to $+60{\;}^{{}^{\circ}}$C) and pressures (from 20 to 600 bar) are given in work [17] and are approximated by the LSM. We found a misprint in the values of the coefficients: the calculated data differed from the measured ones by more than 10%, while according to the authors, the difference should be less than 1%. We used the following approximating function for the sound velocity *V*:(8)$$V^{2}=\frac{\sum_{j=0}^{3}\sum_{k=0}^{2}a_{jk}{\left(p-p_{0}\right)}^{j}{\left(t-t_{0}\right)}^{k}}{\sum_{l=0}^{3}\sum_{m=0}^{2}b_{lm}{\left(p-p_{0}\right)}^{l}{\left(t-t_{0}\right)}^{m}},$$
where $p_{0}=10$ MPa and $t_{0}=250$ K; the experimental data V(p,t) were taken from work [17]: *p* is in (MPa), *t* is in (K) and *V* is in (m/s).

The initial coefficient values for the LSM were taken from work [17]. The adjusted values of the coefficients $a_{jk}$ and $b_{lm}$ are followed:(9)$$a_{jk}=\left(\begin{array}{cccc} 258\cdot 10^{3} & 40.6\cdot 10^{3} & 1.83\cdot 10^{3} & 20.9\\ -4.29\cdot 10^{3} & -382 & 8.58 & 0.549\\ 20.1 & 0.715 & -0.104 & 1.53\cdot 10^{-3} \end{array}\right),$$
(10)$$b_{jk}=\left(\begin{array}{cccc} 1 & 0.130 & 3.73\cdot 10^{-3} & -0.513\cdot 10^{-6}\\ -2.07\cdot 10^{-3} & 401\cdot 10^{-6} & 73.5\cdot 10^{-6} & 0.476\cdot 10^{-6}\\ -14.2\cdot 10^{-6} & 1.17\cdot 10^{-6} & 0.415\cdot 10^{-6} & 44.0\cdot 10^{-12} \end{array}\right).$$

The Table 3 summarizes the physical properties of liquefied SF${}_{6}$ under the conditions that can be realized in experiment.

The AO interaction is resonant in nature and, at a given ultrasound frequency *F*, the AO diffraction is observed only when the radiation is incident on the AO cell at angles θ close to the Bragg angle $\theta_{B}=\lambda F/V$. Similarly, when the radiation is incident on the AO cell at the Bragg angle, the diffracted beam is observed only for a narrow ultrasound frequency band. The resonant frequency $F_{\mathrm{res}}$ is determined by the thickness *d* of the sound transducer: $F_{\mathrm{res}}=V_{\mathrm{PZT}}/2d$ [18]. Here $V_{\mathrm{PZT}}$ is the sound velocity in the material of the sound transducer. The dependence of the AO diffraction efficiency ξ on θ and *F* has the form of the squared sinc-function [19], and the two-sided interaction bandwidths (−3 dB criterion for the diffraction efficiency) can be calculated using the relations [20]:(11)$$\Delta \theta =\frac{0.9nV}{FL_{\mathrm{eff}}},\;\;\;\;\Delta F=\frac{1.8nV^{2}}{\lambda FL_{\mathrm{eff}}},$$
where $L_{\mathrm{eff}}$ is the effective length of the AO interaction region, which is usually equal to or slightly shorter than the length of the sound transducer *L*.

## 3. Results and Discussion

It was found that at the room temperature of +23 ${}^{{}^{\circ}}$C, the diffraction efficiency ξ was proportional to the input electric power $P_{\mathrm{el}}$, while the angular dependence of ξ was well fitted by the squared sinc-function, as was predicted by the theory (see Figure 3a,b). However, the frequency dependence of ξ turned out to be asymmetric. We suppose that this fact was related to the changes in the voltage amplitude *U* with the frequency due to the resonance character of the sound transducer electric properties (Figure 3c). After normalization of the diffraction efficiency to 1 W of the input electric power, the dependence $\xi_{\mathrm{norm}}\left(F\right)$ became symmetric and well-fitted by the squared sinc-function (Figure 3d). The same dependencies measured at the lower temperature $t=10{\;}^{{}^{\circ}}$C are shown in Figure 4. The experimental results for both temperature regimes are summarized in Table 4.

We also calculated properties of the AO modulator using relations given in the previous section (see Table 5).

The angular bandwidth Δθ of the AO modulator is $0.1^{{}^{\circ}}$ greater than that predicted by the theory. We attribute this to the divergence of radiation and ultrasound, which is not taken into account in the model. At the same time, the frequency bandwidth ΔF coincided with theoretical predictions at +23 ${}^{{}^{\circ}}$C within the error. At the temperature of +10 ${}^{{}^{\circ}}$C, the frequency resonance narrowed significantly and side lobes appeared. In our assessment, this fact may be associated with the complex acoustical modes of the transducer at this temperature.

We found that the resonant frequency $F_{\mathrm{res}}$ decreased with increase in the temperature *t* (see Figure 3d and Figure 4d). This trend is confirmed by the results of numerical simulations in work [14]. There are several reasons for this: (1) thermal expansion of the piezoelectric material; (2) decrease in the sound velocity with increase in the temperature; (3) temperature dependence of the sound transducer impedance *Z*. The first and the second reasons directly affected the resonant frequency: $F_{\mathrm{res}}=V_{\mathrm{PZT}}/2d$. The last reason was indirect because the shift of the resonant frequency was affected by change in the electric power $P_{\mathrm{el}}$ (see Equation (Equation 2) and Figure 3c and Figure 4c).

The diffraction efficiencies ξ at the same voltage amplitude *U* at +23 ${}^{{}^{\circ}}$C and +10 ${}^{{}^{\circ}}$C differed by one order. However, only this could not mean that at a temperature of +23 ${}^{{}^{\circ}}$C, the AO modulator performed an order of magnitude better than at +10 ${}^{{}^{\circ}}$C. As can be seen from Figure 3c and Figure 4c, the electrical impedance *Z* of the sound transducer resonantly depended on the frequency *F* and varies with the temperature *t*. Therefore, it was necessary to compare the diffraction efficiency not at the same voltage amplitude *U*, but at the same electric power $P_{\mathrm{el}}$. Unfortunately, we could not make a comparison with data of other researchers since in their works only the dependencies ξ(U) of the diffraction efficiency on the voltage amplitude are given and there are no data on the frequency dependence of the electrical impedance Z(F) of the ultrasound transducer.

The experimental value $\xi_{\mathrm{norm}}\approx 0.23$ (%/W) of the normalized diffraction efficiency was the same for +23 ${}^{{}^{\circ}}$C and +10 ${}^{{}^{\circ}}$C within the error, as predicted by the theory. However, the value of the AO diffraction efficiency achieved in the experiment was less than the value $\xi_{\mathrm{norm}}\approx 0.35$ (%/W), predicted by the theory. Therefore, we introduced a correction coefficient κ≈0.7 that was an attempt to fit the theory to the experimental results. This coefficient was not a universal constant and depended on the material of the ultrasound transducer and its dimensions, as well as on temperature, pressure and ultrasound frequency. There were also a number of factors that determined the value of this correction factor:(1)Relation (Equation 3) assumed that the radiation beam had a plane wavefront, although in fact there was diffraction divergence. However, as the THz radiation beam used in the experiments was about 10 mm wide [21], the divergence could be neglected ($\lambda /d=130\;\mu m/10\;\mathrm{mm}\approx 1^{{}^{\circ}}$).(2)The same applied to the sound beam. However, due to the low sound velocity (V≈300 m/s), the diffraction divergence of the sound beam was significant ($\lambda /d=V/Fd\approx 4^{{}^{\circ}}$) at the frequency F=300 kHz. Because of strong attenuation ($\alpha_{s}\propto F^{2}$), it is not advisable to use ultrasound with a higher frequency for reducing the divergence [9].(3)Only part of the electric power was converted into the acoustic power, because of losses [22] and significant difference in the acoustic impedances of the ultrasound transducer material and the liquefied gas [23].(4)Since the width of the ultrasound transducer was only two times greater than its thickness, a number of acoustic modes with complex structures appeared [24], which was not taken into account in Equation (Equation 3). As a result, the radiation effectively interacted with only a part of the sound beam. Influence of the phase structure of the ultrasonic beam was also possible. Calculation of the structure of the sound field is very complicated and is beyond the scope of this work.

A few words should be said about the stability of measurements. The temperature was controlled by a thermocouple and the accuracy was $0.5{\;}^{{}^{\circ}}$C. When ultrasound with the power of 1 W was turned on for more than 30 s, the diffraction efficiency fell by about a factor of two and stabilized at this lower level. The thermocouple readings remained the same, since it was located at a distance of 5 cm from the ultrasound transducer. Nevertheless, we assumed that the decrease in the diffraction efficiency was associated with an increase in the ultrasound attenuation due to the heating of the SF${}_{6}$ near the ultrasound transducer. At a power of about 0.1 W, no such effect was observed, which made it possible to measure the dependences of the diffraction efficiency on the ultrasound frequency and the angle of incidence of radiation. When we measured the dependence of the diffraction efficiency on the ultrasound power, it was necessary to turn on the ultrasound transducer only for a short time, in order to then correctly carry out averaging. The measurement error was about 10% due to the instability of the laser source.

## 4. Conclusions

In this study, we investigated the operation of an AO modulator based on liquefied SF${}_{6}$ gas in two temperature regimes and took into account the effect of electrical impedance on its characteristics. The dependences of the diffraction efficiency on the voltage amplitude of the electrical signal, on the angle of incidence of THz radiation, and on the sound frequency were measured and analyzed in detail. The developed analytical model of the AO diffraction of THz radiation in liquefied SF${}_{6}$ gas showed good agreement with the experimental results. The temperature was found to affect not only the optical and acoustic properties of the medium of AO interaction, but also the electrical impedance of the sound transducer. For the first time, the influence of the electrical properties of the sound transducer on the characteristics of AO THz radiation modulator based on liquefied SF${}_{6}$ gas was taken into account. This made it possible to estimate the following characteristics of the monochromatic THz radiation AO modulator more correctly: its angular and frequency bandwidths and the diffraction efficiency per 1 W of the input electric power. Moreover, it enabled us to make a correct comparison of the operation of the AO modulator in different temperature regimes. The ultrasound attenuation of liquefied SF${}_{6}$ gas enables the use of low-frequency ultrasound to observe the AO diffraction. As a result, the thickness of the piezoelectric plate must be comparable to its width, which leads to complex types of vibrations of the sound transducer. Therefore, in future work, we plan to investigate the effect of the ratio between the width and thickness of the sound transducer on the characteristics of the AO modulator based on liquefied SF${}_{6}$ gas. We will also continue the search for optimal operating conditions for the AO modulator of THz radiation. Although, according to the theory, high temperatures are more preferable, they lead to decrease in the cavitation threshold and transparency of liquefied SF_6_. Therefore, we are confident that the AO modulator will have the best performance at lower temperatures.

## Figures and Tables

**Figure 1 materials-14-05519-f001:**
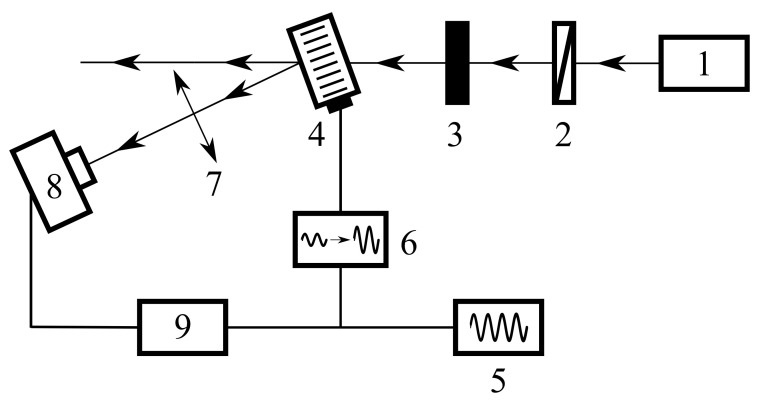
Schematic diagram of experimental setup: 1—FEL; 2—wire-grid polarizer; 3—set of calibrated radiation attenuators; 4—AO cell; 5—signal generator; 6—electrical amplifier; 7—lens; 8—Golay cell; 9—lock-in amplifier.

**Figure 2 materials-14-05519-f002:**
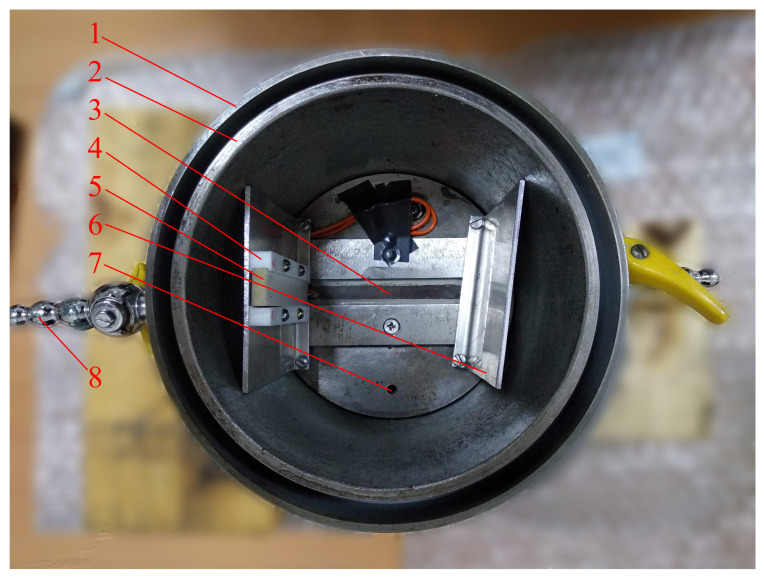
Inside view of AO cell: 1—external wall; 2—internal wall; 3—optical window; 4—sound transducer holder; 5—sound transducer; 6—sound reflector; 7—SF${}_{6}$ gas inlet/outlet; 8—cooling water inlet/outlet.

**Figure 3 materials-14-05519-f003:**
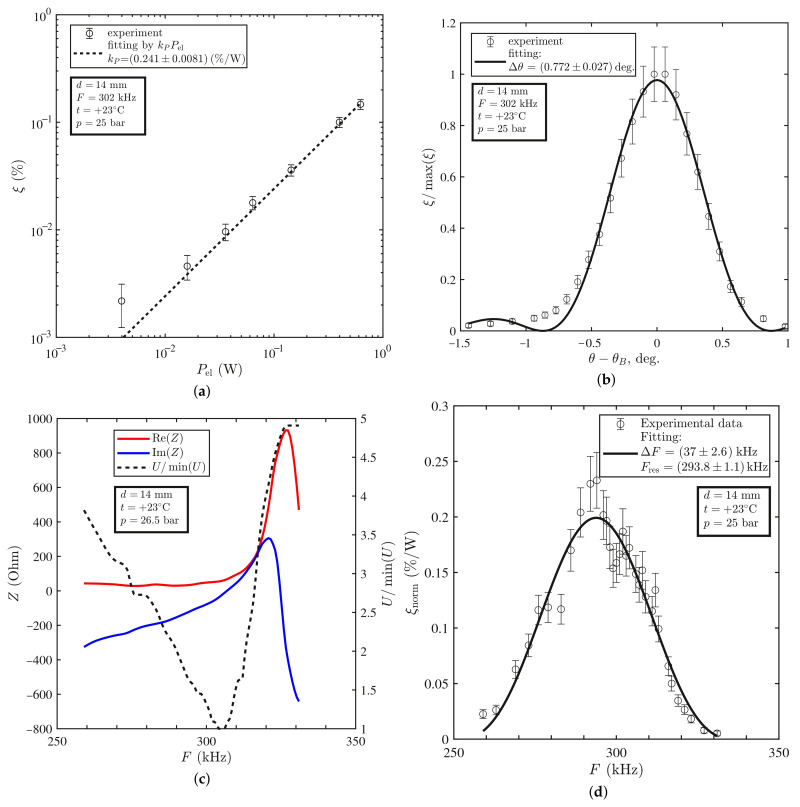
Experimental results obtained at +23 ${}^{{}^{\circ}}$C: (**a**) AO diffraction efficiency ξ vs. input electric power $P_{\mathrm{el}}$; (**b**) AO diffraction efficiency ξ/max(ξ) vs. difference between angle θ of incidence of THz radiation on AO cell and Bragg angle $\theta_{B}$; (**c**) frequency dependences of real and imaginary parts of complex impedance *Z* of sound transducer and amplitude *U* of electrical signal; (**d**) frequency dependence of AO diffraction efficiency per 1 W of input electric power.

**Figure 4 materials-14-05519-f004:**
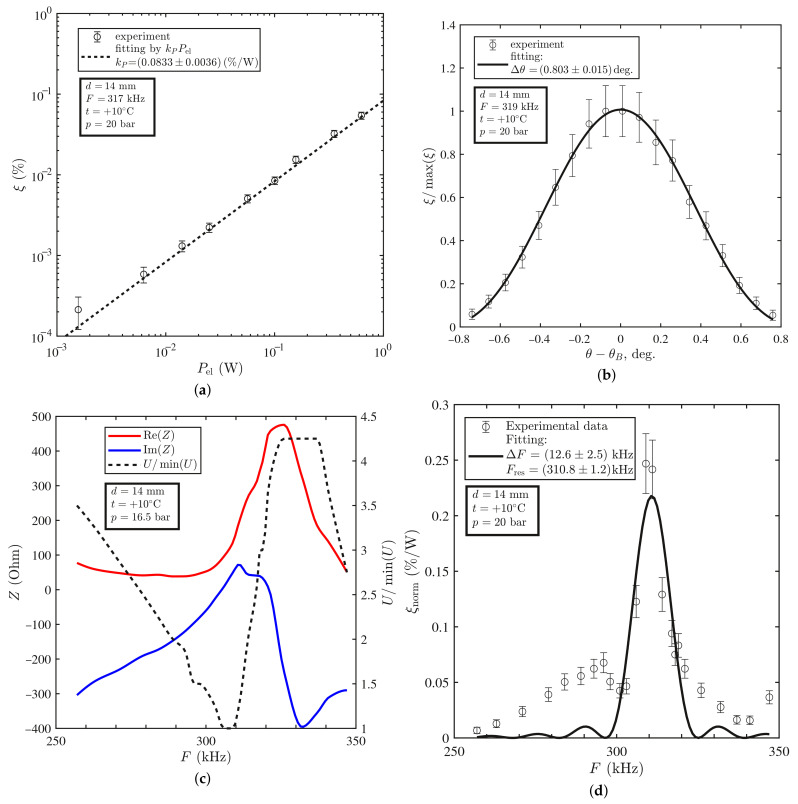
Experimental results obtained at +10 ${}^{{}^{\circ}}$C: (**a**) AO diffraction efficiency ξ vs. input electric power $P_{\mathrm{el}}$; (**b**) AO diffraction efficiency ξ/max(ξ) vs. difference between angle θ of incidence of THz radiation on AO cell and Bragg angle $\theta_{B}$; (**c**) frequency dependences of real and imaginary parts of complex impedance *Z* of sound transducer and amplitude *U* of electrical signal; (**d**) frequency dependence of AO diffraction efficiency per 1 W of input electric power.

**Table 1 materials-14-05519-t001:** Parameters of experimental setup.

λ (μm)	*L* (mm)	*d* (mm)	*l* (cm)
130	80	14	5

**Table 2 materials-14-05519-t002:** Numerical values of coefficients $a_{n}$ in dependence of density ρ on pressure *p* and temperature *t*.

*p* (bar)	$a_{1}$	$a_{2}$	$a_{3}$	$a_{4}\cdot 10^{3}$	$a_{5}\cdot 10^{3}$
20	1570.47	−25.6846	0	0	−11.8502
25	1577.29	33.0886	−0.210490	−4.73339	25.3988

**Table 3 materials-14-05519-t003:** Properties of liquefied SF${}_{6}$ gas.

*t* (${}^{{}^{\circ}}$C)	*p* (bar)	ρ (kg/m${}^{3}$)	*n*	*V* (m/s)	$\alpha_{s}$ (cm${}^{-1}$)	$M_{2}$ ($10^{-15}$ s${}^{3}$/kg)
+23	25	1369.4	1.2362	227	0.40	15,720
+10	20	1490.2	1.2585	283	0.30	9130

**Table 4 materials-14-05519-t004:** Properties of AO modulator of THz radiation revealed in the experiment.

*t* (°C)	*p* (bar)	$k_{U}$ (1/kV^2^)	$\xi_{\mathrm{norm}}$ (%/W)	Δθ (deg)	ΔF (kHz)	$F_{\mathrm{res}}$ (kHz)
+23	25	9.6±0.3	0.23±0.02	0.77±0.03	37±3	294±1
+10	20	1.3±0.1	0.24±0.02	0.80±0.02	12±3	311±1

**Table 5 materials-14-05519-t005:** Properties of AO modulator of THz radiation predicted by the theory.

*t* (${}^{{}^{\circ}}$C)	*p* (bar)	$k_{U}$ (1/kV${}^{2}$)	$\xi_{\mathrm{norm}}$ (%/W)	Δθ (deg)	ΔF (kHz)
+23	25	14.6	0.35	0.62	38
+10	20	1.8	0.33	0.74	56

## Data Availability

The data presented in this study are available on request from the corresponding author. The data are not publicly available due to the further research.

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
