# Peer review of "Temperature Effects in an Acousto-Optic Modulator of Terahertz Radiation Based on Liquefied SF6 Gas"

_materials, 2021, doi:10.3390/ma14195519_

Round 1

Reviewer 1 Report

The authors investigated the operation of an AO modulator based on liquefied SF6 gas in two temperature regimes. Overall, the quality of the presentation is acceptable. I recommend the work can be published in MDPI (materials) provided the following few points are properly incorporated into the manuscript.

  1. While the experimental work focused on two temperatures of 10C and 23 C, in abstract line 3, the authors addressed findings in the range of +10C to +24C. Please double-check the provided numbers in the entire text.
  2. In the introduction, it should be better explained what is the necessity of using AO in THz range. Some examples with Refs will be helpful.
  3. The motivation of the work should be better explained in the introduction. Authors should better explain the importance of the experimental study of temperature effects in AO modulators in THz.
  4. Writing in few sentences seems confusing for readers and I suggest considering re-writing them. Examples include lines 42-43-44.

Author Response

Point 1

While the experimental work focused on two temperatures of 10C and 23 C, in abstract line 3, the authors addressed findings in the range of +10C to +24C. Please double-check the provided numbers in the entire text.

Response 1

Thanks for helping me find this typo. The correct temperature value is 23 degrees Celsius. The abstract has been corrected (page 1, line 3).

========================================

Point 2

In the introduction, it should be better explained what is the necessity of using AO in THz range. Some examples with Refs will be helpful.

Response 2

The introduction has been expanded. The factors showing the prospects of using AO devices in the THz range are presented (page 1, lines 18-21,36; page 2, lines 37-42).

========================================

Point 3

The motivation of the work should be better explained in the introduction. Authors should better explain the importance of the experimental study of temperature effects in AO modulators in THz.

Response 3

The Introduction was expanded by the motivation part (page 2, lines 72-74).

========================================

Point 4

Writing in few sentences seems confusing for readers and I suggest considering re-writing them. Examples include lines 42-43-44.

Response 4

The passage has been rewritten (page 2, lines 48-53).

Reviewer 2 Report

This paper is devoted to the investigation of AO interaction in THz frequency range. It is well structured and represents mainly the experimental results. Nevertheless, the work has a number of serious drawbacks.

In the Introduction:

  1. The connection between the following sentences is not clear: "Unfortunately, ultrasonic attenuation is large in the gases even under high pressure. That is why experiments carried out in [7] with various inert gases in the liquid phase obtained the AO diffraction efficiency of about 80%." The 80% diffraction efficiency is very high even for IR radiation.
  2. "To do this, we have recently developed a model for the AO diffraction of THz radiation in the liquefied gas, which takes into account the effect of temperature on its physical properties [9]. However, this model failed to explain the trend revealed in [7] and even on the contrary predicted increase in the diffraction efficiency with increasing temperature in the range from +10 to +30◦C at an ultrasound frequency of 300 kHz."  I consider that authors should not discuss here the model that failed to explain experimental results. Could you also clarify the choice of 300kHz ultrasound frequency?
  3.  What does it mean - transducer thickness 6 mm at 300kHz?
  4. I also consider that most part of introduction is devoted to the discussion of questions that are not directly related to the main topic of the paper.

In the Section 2:

  1. Line 74 - It is not correct to say that AO cell was made from steel.
  2. "Then the ultrasound frequency was taken equal to the resonance frequency of the sound transducer and the dependence of the intensity I1 on the angle q of the AO cell turn was measured." Please clarify this statement as the transducer resonant frequency varies with temperature.
  3. The general comment for this section is that it seems useless to discuss the dependencies of AO diffraction characteristics obtained for fixed voltage as it says nothing about the power of acoustic wave aroused in the AO interaction media. Especially in the case of standing wave ratio (SWR) strongly dependent on frequency. So the authors may remove from the paper all such information and figures 3c and 4c. Fig. 3a and 3c should be recalculated into the dependencies of diffraction efficiency on power.
  4. Table 5 should be transferred to Section 2 after the equations describing the SF6 physical properties. Is it possible to compensate the pressure variation with temperature? It seems that pressure variation is mostly responsible for the change of SF6 physical properties.

    Section 3:

  1. The frequency axis in Fig.3c-e and Fig.4c-e does not allow to define anything from the presented graphs. Please increase the number of numeric values on the frequency axis.
  2. It is more informative to show not the frequency dependences of the impedance, but the frequency dependence of the SWR
  3. It seems that the observed temperature effect may be explained by the joint action of SWR temperature dependence (due to the thermal expansion of the transducer) and pressure variation.
  4. The introduction of correction coefficient looks like a rough attempt to fit the theory to the experimental results. It is also obvious that this coefficient should depend on frequency

Author Response

>>> In Introduction <<<

Point 1

The connection between the following sentences is not clear: "Unfortunately, ultrasonic attenuation is large in the gases even under high pressure. That is why experiments carried out in [7] with various inert gases in the liquid phase obtained the AO diffraction efficiency of about 80%." The 80% diffraction efficiency is very high even for IR radiation.

Response 1

The sentences have been rewritten (page 1, lines 29-32).

================================

Point 2

"To do this, we have recently developed a model for the AO diffraction of THz radiation in the liquefied gas, which takes into account the effect of temperature on its physical properties [9]. However, this model failed to explain the trend revealed in [7] and even on the contrary predicted increase in the diffraction efficiency with increasing temperature in the range from +10 to +30◦C at an ultrasound frequency of 300 kHz."  I consider that authors should not discuss here the model that failed to explain experimental results. Could you also clarify the choice of 300kHz ultrasound frequency?

Response 2

1) The discussion of the model was removed. (page 2, lines 46-48)

2) The choice of a frequency of 300 kHz is due to the need to deflect the diffracted beam at an angle much larger than the divergence angle of this beam, which is about 1 degree for a beam diameter of 10 mm and a wavelength of 130 μm. At an ultrasound frequency of 300 kHz, the deflection angle is about 8 degrees, which is sufficient for its spatial separation from the transmitted radiation beam. The use of higher frequencies is not advisable because of the high attenuation of ultrasound, which is proportional to the square of the frequency. (page 2, lines 55-61)

================================

Point 3

What does it mean - transducer thickness 6 mm at 300kHz?

Response 3

The resonant frequency F0 is determined by the thickness h of the transducer and sound velocity V: F0 = 2h/V. V is about 3 km/s. Therefore h = V/2F = 5 mm. The material of the transducer in our experiment had slightly lower sound velocity, that is why h was 6 mm. (page 2, lines 62-65)

================================

Point 4

I also consider that most part of introduction is devoted to the discussion of questions that are not directly related to the main topic of the paper.

Response 4

The introduction to the article has been revised.

================================

>>>> In Section 2 <<<<

Point 1

Line 74 - It is not correct to say that AO cell was made from steel.

Response 1

The sentence has been corrected (page 3, lines 88-89)

===================================

Point 2

"Then the ultrasound frequency was taken equal to the resonance frequency of the sound transducer and the dependence of the intensity I1 on the angle q of the AO cell turn was measured." Please clarify this statement as the transducer resonant frequency varies with temperature.

Response 2

The transducer resonant frequency depends on the temperature. Therefore, at a given temperature, we chose the optimal frequency at which the AO diffraction efficiency reached its maximum value. (page 3, lines 104-106)

===================================

Point 3

The general comment for this section is that it seems useless to discuss the dependencies of AO diffraction characteristics obtained for fixed voltage as it says nothing about the power of acoustic wave aroused in the AO interaction media. Especially in the case of standing wave ratio (SWR) strongly dependent on frequency. So the authors may remove from the paper all such information and figures 3c and 4c. Fig. 3a and 3c should be recalculated into the dependencies of diffraction efficiency on power.

Response 3

Figures 3c and 4c have been removed. Fig. 3a and 4a have been recalculated.

===================================

Point 4

Table 5 should be transferred to Section 2 after the equations describing the SF6 physical properties. Is it possible to compensate the pressure variation with temperature? It seems that pressure variation is mostly responsible for the change of SF6 physical properties.

Response 4

Table 5 was transferred to Section 2 after the equation. (page 5, lines 139-140 and top of the page 6).

Yes, changes in pressure with temperature can be compensated for by venting some of the gas from the cuvette. Based on the physical properties of SF6, we can say that they depend on both pressure and temperature. Most likely, it is also important how close the conditions are to the critical state. Therefore, it is advisable to carry out additional investigations by fixing one of the parameters, i.e. either at constant temperature or at constant pressure.

===================================

>>>> In Section 3 <<<<

Point 1

The frequency axis in Fig.3c-e and Fig.4c-e does not allow to define anything from the presented graphs. Please increase the number of numeric values on the frequency axis.

Response 1

The number of ticks on the frequency axis in Figures has been increased.

===================================

Point 2

It is more informative to show not the frequency dependences of the impedance, but the frequency dependence of the SWR.

Response 2

From a consumer point of view, you are right, because SWR characterizes energy consumption. However, together, the dependences of the real and imaginary parts of the electrical impedance on frequency are more informative and make it possible to reveal the dynamics of the temperature dependence of the emitter impedance on temperature for various ultrasound emitters. This study is planned for the near future.

===================================

Point 3

It seems that the observed temperature effect may be explained by the joint action of SWR temperature dependence (due to the thermal expansion of the transducer) and pressure variation.

Response 3

The influence of the SVR(t), indeed, must be taken into account, as well as the temperature dependence of the transducer thickness. Nevertheless, there are still a number of factors that affect the operation of the AO modulator. These factors are summarized at the end of Section 3.

===================================

Point 4

The introduction of correction coefficient looks like a rough attempt to fit the theory to the experimental results. It is also obvious that this coefficient should depend on frequency.

Response 4

You're right. The value of this coefficient is not a universal constant and depends on the material of the ultrasound transducer and its dimensions, as well as on temperature, pressure and frequency. (page 9, lines 186-190)

Reviewer 3 Report

The manuscript of Dr. Nikitin et al. should be considered as the next step in optimization of a novel acousto-optic modulator based on liquefied gas. The authors considered effects of temperature on the diffraction efficiency of the AO device under investigation. The authors performed a series of measurements for 2 temperature levels, and extended their initial theoretical model of AO interaction. Generally, the manuscript is well written, which helps a reader to understand the topic. The authors showed state-of-the-art and motivated their study thoroughly. The aim of the work was given precisely. The results are described thoroughly with a short discussion. However, there are some issues / questions that could be addressed:

(1) p. 1, line 17 – is ‘created’, should be ‘developed’

(2) p. 2, line 42 – is ‘it does not take …’, should be ‘it did not take …’ (change the tense)

(3) p. 2, line 68 – is ‘turned through Bragg angle’, should be ‘was tilted at the Bragg angle…’ (please, make sure the sentence is clear)

(4) Discussion of the results should be definitely expanded. Please, provide information on the stability of the measurments (e.g. to which degree were you able to control the temperature, how much time did the measurements take?)

(5) p. 3, line 85 – please specify the details of the method of successive approximation. E.g. how did you obtain Eq. 3 ??

(6) Please, also specify clearly where is the advance in theoretical calculations (which aspect you modified?)

(7) Why those 2 temperature levels were selected?

Author Response

Point 1

p. 1, line 17 - is 'created', should be 'developed'

Response 1

the word 'created' has been replaced by 'modern'

================================

Point 2

p. 2, line 42 - is 'it does not take ...', should be 'it did not take ...' (change the tense)

Response 2

This sentence has been removed from the text.

================================

Point 3

p. 2, line 68 - is 'turned through Bragg angle', should be 'was tilted at the Bragg angle ...' (please, make sure the sentence is clear)

Response 3

'turned through Bragg angle' has been replaced by 'was tilted at the Bragg angle' (p. 2, lines 81-82)

================================

Point 4

Discussion of the results should be definitely expanded. Please, provide information on the stability of the measurments (e.g. to which degree were you able to control the temperature, how much time did the measurements take?)

Response 4

The Discussion has been expanded (page 9, lines 209-214; page 10, lines 215-222)

================================

Point 5

p. 3, line 85 - please specify the details of the method of successive approximation. E.g. how did you obtain Eq. 3 ??

Response 5

The sentence has been rewritten. (page 3, lines 99-102)

Equation (3) was taken from [11] ([9] in the original version). It is a well-known relationship for diffraction efficiency, but in which the attenuation of the ultrasonic beam is additionally taken into account in the form of an exponential factor.

=================================

Point 6

Please, also specify clearly where is the advance in theoretical calculations (which aspect you modified?)

Response 6

There is no significant novelty in theoretical calculations. Our contribution is that we carried out a literature review of the physical properties of SF6 and presented analytical dependences of density, sound speed, ultrasound attenuation coefficient on temperature and pressure for this substance. In comparison with Durr's work, we additionally took into account the influence of the electrical impedance of the ultrasound emitter, and also determined the dependence of the impedance on temperature and pressure. This allowed us to determine the efficiency of diffraction per 1 W of electrical power and to make a correct comparison of the operation of the AO modulator under different conditions.

=================================

Point 7

Why those 2 temperature levels were selected?

Responce 7

As the temperature +23C is approximately the same as in our previous work [11] ([9] in original version), and +10C is approximately the same as in Durr's work [9] ([7] in original version).

Round 2

Reviewer 2 Report

Even with the improvements made by authors, I consider that the experimental methodology and interpretation of the results obtained raise doubts about their correctness and the study itself cannot be considered as complete.